Diversity analysis of microorganisms on the surface of four summer fruit varieties in Baotou, Inner Mongolia, China

He Shan 1
Gao Li 2
Zhang Zhuomin 2
Ming Zhihui 2
Gao Fang 1
Ma Shuyi 1 mashuyi-2008@163.com
Zou Mingxin 2 zoumingxin@126.com
1 School of Medical Technology and Anesthesiology, Baotou Medical College , Baotou , China
2 Faculty of Ecology and Environment, Baotou Teacher’s College , Baotou , China
Thomas Jonathan
Electronic publication date: 2024 Dec 19
Publication date: 2024
Volume: 12
Electronic Location ID: e18752
Received 2024 Jul 12; Accepted 2024 Dec 3
Copyright: © 2024 He et al.
Copyright year: 2024
Copyright holder: He et al.
License: This is an open access article distributed under the terms of the Creative Commons Attribution License, which permits unrestricted use, distribution, reproduction and adaptation in any medium and for any purpose provided that it is properly attributed. For attribution, the original author(s), title, publication source (PeerJ) and either DOI or URL of the article must be cited.
License URL: https://creativecommons.org/licenses/by/4.0/

Keywords: Fruit surfaces, Bacteria, Fungi, Correlation

Funding: Youth Program of Baotou Medical College BYJJ-ZRQM202314 Baotou Medical College Innovation Team Project bycxtd-14 Inner Mongolia Autonomous Region level College Student Innovation and Entrepreneurship Plan Training Project S202410131030 This work was funded by the Youth Program of Baotou Medical College, grant number BYJJ-ZRQM202314; the Baotou Medical College Innovation Team Project, grant number bycxtd-14; and the Inner Mongolia Autonomous Region level College Student Innovation and Entrepreneurship Plan Training Project, grant number S202410131030. The funders had no role in study design, data collection and analysis, decision to publish, or preparation of the manuscript.

==============================
Improper storage of post-harvest fruits leads to significant losses, especially due to microbial-induced decay. Understanding the naturally occurring microbial communities on fruit surfaces and their functions is the first step in the development of new strategies for controlling post-harvest fruit decay. These new strategies could generate significant economic value by improving fruit preservation and extending the shelf-life of fruit. In the present study, 16S rRNA and ITS high-throughput sequencing technologies were used to analyze the diversity and composition of microorganisms on the surfaces of four different fruit varieties: three plum varieties and one apple variety, all from the same orchard in Donghe District, Baotou City, China. The results displayed no notable difference in bacterial diversity on the surfaces of the four varieties of fruits (P > 0.05), but there were significant differences in fungal diversity (P < 0.05). The most abundant bacterial phyla detected on the fruit surfaces were Proteobacteria, Bacteroidota, and Firmicutes; the most abundant fungal phyla were Ascomycota, Basidiomycota, and Mortierellomycota. Though microbial compositions on the fruit surfaces differed between the fruits, the surface microbial community structure of the three plum varieties exhibited higher similarity, indicating that fruit type is a key factor influencing the composition of surface microorganisms. There were also differences in the epidermal microbial community composition between the fruits involved in this study and fruits of the same species reported from other regions, suggesting that geographical factors also play a critical role in microbial composition. The correlation analysis revealed significant associations between the microorganisms with the highest abundance on the surface of the fruits, suggesting the existence of symbiotic and mutualistic relationships between these microorganisms, but the specific mechanisms behind these relationships need to be further explored. This study provides a basis for the establishment of post-harvest fruit preservation strategies.

Introduction

Post-harvest fruits have reduced defense capabilities because they are no longer receiving nutrients from the parent plant. Without appropriate measures, these fruits are easily infected by decay-related microorganisms, leading to substantial loss. Furthermore, both because of respiration and to maintain normal physiological activity, post-harvest fruits consume a significant amount of nutrients, which markedly shortens their shelf life (Zhang, Xin & Xu, 2014).

To prevent post-harvest fruit losses, various preservation technologies have emerged. Common preservation techniques can be categorized into physical, chemical, and biological preservation (Zhu et al., 2014). Biological preservation mainly involves using biological preservatives to extend the storage life of food. Bio-preservatives are mainly derived from plants, animals, and microorganisms. Because of its biological origins, bio-preservation technology has the advantages of being non-toxic, harmless, non-residual, non-resistant, and easy to degrade. For these reasons, bio-preservation has become an effective way to achieve pollution-free preservation of fruits and vegetables and is one of the developmental directions for new biological preservation technologies (Zhang et al., 2018b; Wang, Hu & Hu, 2005; Xu & Zhang, 2018). Several post-harvest biological control products have been developed based on single antagonistic agents. The preservation mechanism of antagonistic bacteria involves producing antibacterial substances like antibiotics, bacteriocins, and lysozymes. These antagonistic microorganisms can inhibit harmful microorganisms in fruits or compete with them for carbohydrates and other nutrients in fruits, preventing the decline of vitamin C, sugar content, and SOD activity during storage, thus preventing fruit rot and keeping the fruit fresh (Bencheqroun et al., 2006).

Microbial agents have been effectively applied in fruit preservation, with numerous successful cases. For instance, lactic acid bacteria (LAB) can prevent the growth of Salmonella typhimurium, Escherichia coli, and Listeria monocytogenes in lettuce and apples (Agriopoulou et al., 2020; Amiri et al., 2021). The fermentation broth of Streptomyces strains effectively prevents post-harvest rot in wax apple and guava fruits, helping them maintain their post-harvest quality (Bai et al., 2022). Bacteriocin from Lactobacillus brevis SM6 shows excellent preservation effects on ready-to-eat apricots (Dharwal et al., 2020). Lactobacillus plantarum TPB21.12 combined with maize starch and kappa-carrageenan can effectively preserve sliced apples, extending their shelf life (Kusnadi et al., 2023), and Lactobacillus plantarum along with konjac glucomannan has been shown to effectively prolong the shelf life of fresh-cut kiwifruit (Hashemi & Jafarpour, 2021).

However, microbial preservation technology is still in the early research stages, and faces significant limitations, such as a high sensitivity to environmental factors and the high costs of some bacteriocins. Research indicates that the structure of microbial communities on fruit surfaces is influenced by both fruit type and geographical conditions such as altitude, latitude, and longitude (Gao et al., 2019). A previous study showed that the microbial communities on grape surfaces are dominated by Proteobacteria, with fungi primarily belonging to the phylum Ascomycota (He et al., 2022). On apple surfaces, the dominant bacteria belong to the phylum Actinobacteriota (34.57%), and the primary fungi are also from the phylum Ascomycota (Zhang et al., 2022). The microbial communities on mango surfaces are predominantly Burkholderiaceae bacteria (58.8%) and fungi from the Sordariaceae (34.5%), Chaetomiaceae (23.21%), and Fabaceae (13.09%) families (Taîbi et al., 2021), indicating that different fruits have distinct microbial communities, and that the microbial compositions of even the same type of fruit can vary by geographic region. Therefore, the microbial antagonists used for preservation should also be distinct. Understanding the naturally occurring microbial communities on fruit surfaces and their functions can help provide the basis needed to establish effective control strategies for postharvest spoilage of specific fruit types in specific regions. This type of research also helps identify natural microbial antagonists, which have significant economic value because of their role in fruit preservation and storage.

This study investigated the microbial diversity and functions of the microbes on the surfaces of one variety of apple and three varieties of plum grown in Baotou, China: Summer Red Apple, (XHPG), Yu Emperor Plum (YHL), Crystal Sugar Plum (BTL), and Yu Plum (YUL). Both 16S rRNA and ITS high-throughput sequencing technologies were used to analyze the diversity and functional relationships of surface microorganisms, providing a research basis for post-harvest fruit preservation.

Materials and Methods

Sample collection

In July 2023, samples were collected from the Wang Dahan Village (110°00′36.79″E, 40°32′49.07″N), Donghe District, Baotou City, Inner Mongolia, China. One apple variety (XHPG) and three plum varieties (YHL, BTL, and YUL) were planted in the same orchard with identical environmental conditions, so these four varieties were selected to study differences in the surface microbiota of different types of fruits grown under the same environmental conditions. Fresh, plump, disease-free fruits were collected from three trees using scissors sterilized with 75% alcohol. The fruits were placed in sterile, self-sealing bags and then transported to the laboratory under cryogenic conditions. In the lab, fruit surfaces were collected using sterilized scissors, and then washed with 200 mL PBS buffer (0.1 mol/L, pH 7.0), vortexed thoroughly at 200 r/min for 30 min, ultrasonicated for 15 min, and filtered through a 0.22 μm microporous membrane. The membranes were then cut with sterile scissors, placed in sterile centrifuge tubes, and stored at −80 °C for further analysis.

DNA extraction, PCR amplification, and high-throughput sequencing

DNA was extracted using a Magbead Soil And Stool DNA Kit (Cwbio, Shanghai, China), according to the manufacturer’s instructions. The quality of the extracted DNA was detected using 1% agarose gel electrophoresis. Primers 338F (5′-ACTCCTACGGGAGGCAGCA-3′) and 806R (5′-GGACTACHVGGGTWTCTAAT-3′) were used to amplify the V3–V4 hypervariable regions of the bacterial 16S rRNA gene. The length of the PCR product was 469 bp. Primers ITS1F (5′-CTTGGTCATTTAGAGGAAGTAA-3′) and ITS2R (5′-GCTGCGTTCTTCATCGATGC-3′) were used to amplify the fungal ITS gene region. The length of the PCR product was between 163 and 447 bp. The PCR reaction procedure was as follows: pre-denaturation at 94 °C for 5 min; denaturation at 94 °C for 30 s; annealing at 55 °C for 30 s; extension at 72 °C for 45 s, 35 cycles; extension at 72 °C for 10 min; and storage at 4 °C. The purified DNA was sequenced using the Illumina NovaSeq platform by Biomarker Technologies.

Data analysis

The raw data were merged (FLASH, version 1.2.11), and the merged sequences were quality filtered (Trimmomatic, version 0.33) and chimera-checked (UCHIME, version 8.1), resulting in high-quality tag sequences. These sequences were then clustered according to a criterion of 97% similarity (USEARCH, version 10.0). Bacterial species annotation was performed using the Silva 138 database with RDP Classifier software, with a confidence threshold of 0.8. Fungal species annotation was performed using the UNITE 8.0 database with RDP Classifier software, with a confidence threshold of 0.8. Phylogenetic analysis of dominant OTUs was performed using PyNAST; the neighbor-joining method was selected to construct the phylogenetic tree. Alpha diversity analysis was conducted using Mothur (version v.1.30, http://www.mothur.org/). Rarefaction curves were used to evaluate whether the sequencing depth covered all communities in the samples. Beta diversity analysis was performed by calculating unweighted Unifrac distances and binary_jaccard Unifrac distances using Qiime software (Version 1.9.1), drawing the NMDS diagram using the vegan package in the R software (Version 2.15.3), and then using the PERMANOVA test to study the differences in microbial community structure between sample groups. The effect of the abundance of each species on differences between sample groups was analyzed by a linear discriminant effect size (LefSe) analysis; LDA > 4 was used as the standard to screen significantly differentially abundant species. Spearman’s correlations were conducted to prioritize indicator species linking fungi and bacteria, which were visualized using BMKCloud (www.biocloud.net). Bacterial functions were predicted using the Picrust2 (Version 2.3.0) software (Douglas et al., 2020), and fungal functions were predicted using the FUNGuild (Version 1.0) software (Nguyen et al., 2016).

Results

OTU clustering analysis of microbial sequences on different fruit surfaces

Genomic DNA was extracted from 12 fruit peel samples (four varieties of fruits were named as BTL, XHPG, YHL, and YUL; three fruit peel samples were taken from each variety of fruit), and the V3–V4 hypervariable regions of the 16S rRNA gene were amplified and sequenced. A total of 815,379 (60,028–71,135) valid sequences were obtained (the number of OTUs per sample are shown in Table 1). At 97% sequence similarity, 1,779 OTUs were obtained, belonging to 27 phyla, 56 classes, 145 orders, 254 families, and 431 genera. Sequencing of the ITS1 region of the same fruit surfaces resulted in 737,880 (51,247–67,675) valid sequences (the number of OTUs per sample are shown in Table 2). At 97% sequence similarity, 7,784 OTUs were obtained, belonging to 17 phyla, 56 classes, 139 orders, 320 families, and 745 genera.

Table 1 Basic information of 16s rRNA high-throughput sequencing data of microorganisms on the surface of four kinds of fruit.

Sample	Raw reads	Clean reads	Effective
reads	Number of OTUs	Number of taxa of different taxonomic categories	
Phylum	Class	Order	Family	Genus	
BTL1	80,249	71,072	70,839	62	13	16	25	36	38	
BTL2	80,107	71,336	71,135	57	12	15	25	35	38	
BTL3	80,257	71,233	70,131	90	12	19	41	58	68	
XHPG1	79,960	71,033	60,028	233	17	24	47	70	93	
XHPG2	80,098	70,547	63,497	467	20	35	85	132	193	
XHPG3	79,946	69,280	66,893	177	15	27	58	86	100	
YHL1	80,034	71,029	69,918	88	11	15	36	49	58	
YHL2	79,846	70,989	70,017	127	11	19	45	66	83	
YHL3	80,178	71,046	64,300	495	19	33	81	135	192	
YUL1	79,802	70,989	69,542	73	10	17	32	46	53	
YUL2	80,032	70,479	69,809	72	12	17	32	39	43	
YUL3	79,881	70,526	69,270	74	13	16	33	43	51	

Table 2 Basic information of high-throughput sequencing of ITS gene of fungi in four types of fruit surfaces.

Sample	Raw reads	Clean reads	Effective
reads	Number of OTUs	Number of taxa of different taxonomic categories	
Phylum	Class	Order	Family	Genus	
BTL1	79,797	67,795	63,428	1,117	11	38	92	209	388	
BTL2	79,795	68,385	63,554	1,279	14	42	100	216	419	
BTL3	80,249	68,688	63,393	1,268	13	42	103	214	410	
XHPG1	79,980	70,261	67,137	983	13	41	93	192	361	
XHPG2	79,945	70,246	67,675	825	11	36	89	184	317	
XHPG3	80,123	69,620	67,113	853	13	40	85	189	340	
YHL1	63,807	55,103	51,247	755	12	35	81	174	302	
YHL2	79,955	68,452	62,185	716	12	37	82	179	309	
YHL3	79,956	68,002	62,939	741	11	36	83	179	312	
YUL1	80,018	63,276	56,928	1,059	14	38	92	194	353	
YUL2	79,641	63,382	56,766	961	12	37	90	190	329	
YUL3	79,949	63,222	55,515	1,027	12	36	87	183	345	

Alpha diversity analysis

All the rarefaction curves of the 16S rRNA and ITS sequencing results of the 12 samples were flat (Figs. 1A, 1D), demonstrating that microbial diversity did not continue to increase as sequencing depth increased. This suggests that the sequencing data sufficiently reflected the majority of the microbial diversity in the samples. Venn diagrams comparing the bacterial diversity of different fruit surfaces showed eight shared OTUs among the four fruits, with YHL having 606 unique OTUs, YUL having 173 unique OTUs, XHPG having 728 unique OTUs, and BTL having 165 unique OTUs (Fig. 1B). For fungal abundance, 44 OTUs were shared among the four fruits, with YHL having 1,336 unique OTUs, YUL having 1,868 unique OTUs, XHPG having 1,594 unique OTUs, and BTL having 2,480 unique OTUs (Fig. 1E). At the same sequencing depth, the Shannon index analysis of bacterial alpha diversity on different fruit surfaces showed no significant differences (P > 0.05; Fig. 1C). However, the Shannon index analysis of fungal diversity showed significant differences between all the fruits except between XHPG and BTL (P < 0.05; Fig. 1F).

Figure 1 (A–F) Alpha diversity analysis of microbial communities on four fruit surfaces.

Beta diversity analysis

Beta diversity analysis based on unweighted Unifrac NMDS showed clustering trends in the same group of fruit samples, indicating significant differences in microbial community composition between sample groups (stress < 0.2, P < 0.05; Fig. 2A). Beta diversity analysis based on binary_jaccard NMDS for fungal communities also showed clustering trends among samples from the same group and significant separations between the different groups of samples, indicating that the distribution of epidermal fungi significantly differed between different fruit species (stress < 0.2, P < 0.05; Fig. 2B). Thus, the composition of the bacterial and fungal communities on the surfaces of the four varieties of fruit were significantly different (P < 0.05). A clustering analysis of the top 10 most abundant bacterial and fungal genera on different fruit surfaces showed that BTL, YUL, and YHL were closely related, indicating similar bacterial and fungal compositions among these three plums. These results suggest that fruit type is a key factor influencing microbial composition.

Figure 2 (A–D) Beta diversity analysis of microbial communities on four fruit surfaces.

Analysis of fruit epidermal microbial community composition

Further analysis was conducted on the microbial composition on the surfaces of different fruits. Bacteria detected in all samples were distributed among 27 phyla, and the top 10 phyla with the highest bacterial abundance were Proteobacteria, Bacteroidota, Firmicutes, Actinobacteriota, Acidobacteriota, unclassified_Bacteria, Cyanobacteria, Chloroflexi, Myxococcota, and Gemmatimonadota (Fig. 3A). The phylum with the highest abundance across all four samples was Proteobacteria. At the genus level, 431 bacterial genera were identified across the four fruits (Fig. 3B). The bacterial genera with the highest average relative abundance were unclassified_Bacteria, Sphingomonas, unclassified_Cyanobacteriales, Rothia, Massilia, Pantoea, Pseudomonas, unclassified_Gemmatimonadaceae, Hymenobacter, and Escherichia_Shigella. In the BTL sample, the genus with the highest abundance was Sphingomonas. The genus with the highest abundance in the XHPC sample was Pantoea. Unclassified_Bacteria had the highest abundance in both the YHL and YUL samples.

Figure 3 (A–D) Microbial composition of the surfaces of four fruits.

The fungi identified in all samples were distributed among 17 phyla. The fungal phyla with the highest average relative abundance were Ascomycota, Basidiomycota, Mortierellomycota, unclassified_Fungi, Chytridiomycota, Rozellomycota, Glomeromycota, Olpidiomycota, Mucoromycota, and Neocallimastigomycota (Fig. 3C). The phylum with the highest abundance across all four samples was Ascomycota. At the genus level, 745 fungal genera were identified across the samples (Fig. 3D). The fungal genera with the highest average relative abundance were Mortierella, Fusarium, unclassified_Fungi, unclassified_Didymellaceae, Aspergillus, Cladosporium, Alternaria, Filobasidium, Fusicolla, and Talaromyces. The genera with the highest abundance in each of the samples were: Mortierella in the BTS sample, unclassified_Didymellaceae in the XHPG sample, Aspergillus in the YHL sample, and unclassified_Fungi in the YUL sample.

Difference analysis of microbial community between different sample groups

A LEfSe analysis was conducted to analyze the differences in the microbial composition of the fruit surface between samples. As shown in Fig. 4A, at the phylum level, the abundance of Myxococcota in the BTL sample was significantly higher than in the other three groups, and the abundance of Fusobacteriota in the XHPG sample was significantly higher than in the other groups (LDA > 2.0, P < 0.05). At the genus level, the marker bacterial genera in the XHPG sample were Hymenobacter, unclassified_Erwiniaceae, Cetobacterium, Nitrospira, and Curtobacterium; the marker bacterial genera in the YHL sample were unclassified_Enterobacteriaceae and unclassified_Enterobacterales; and the marker bacterial genera in the YUL sample were Allorhizobium, Neorhizobium, Pararhizobium, Rhizobium, and Brevundimonas (LDA > 2.0, P < 0.05).

Figure 4 (A, B) Microbial communities significantly differing between the surfaces of four fruits.

The differences in fungal composition on the surfaces of different fruits were also analyzed by LEfSe. As shown in Fig. 4B, at the phylum level, the abundance of Mortierellomycota in the BTL sample was significantly higher than in the other samples; the abundance of Basidiomycota in the XHPG sample was significantly higher than in the other fruit samples; the abundance of unclassified_Fungi in the YUL sample was significantly higher than in the other fruit samples; and the abundance of Ascomycota in the YHL sample was significantly higher than in the other fruit samples.

At the genus level, the abundance of Tausonia, Pseudeurotium, Arthrographis, Fusarium, unclassified_Agaricomycetes, Mortierella, Gibellulopsis, Fusicolla, and Talaromyces were significantly higher in BTL than in the other fruits; the abundance of Nigrospora, Aureobasidium, Hanseniaspora, Penicillium, Xerochrysium, and Aspergillus were significantly higher in YHL than in the other fruits; the abundance of Alternaria, unclassified_Didymellaceae, Kabatina, Filobasidium, and Cladosporium were significantly higher in XHPG than in the other fruits; and the abundance of unclassified_Fungi, Cryptococcus, Podospora, and unclassified_Microascaceae were significantly higher in YUL than in the other fruits.

Microbial correlation analysis

The correlations between the different bacteria and fungi were analyzed using a network analysis. There were significant correlations among bacteria (Fig. 5A). The top ten pairs of bacteria with the strongest correlations were as follows: MND1 was highly positively correlated with unclassified_Vicinamibacterales and Marivivens (0.9761, 0.9044); Methylocystis was highly positively correlated with Haliangium (0.9442); unclassified_Micromonosporaceae was highly positively correlated with Algoriphagus (0.9384) and highly negatively correlated with unclassified_Bacteria (−0.9147); Algoriphagus was highly negatively correlated with unclassified_Bacteria (−0.9144); Parabacteroides was highly positively correlated with Sphingomonas (0.8656); Marivivens was highly positively correlated with unclassified_Vicinamibacterales (0.8486); unclassified_Acidobacteriales was highly negatively correlated with Bacteroides (−0.8472); and Ellin6067 was highly negatively correlated with Bacteroides (−0.8422). There were also significant correlations among fungi (Fig. 5B). The top ten pairs of fungi with the strongest correlations were as follows: Filobasidium was highly positively correlated with Alternaria (0.9790); Thelebolus was highly negatively correlated with Arthrographis and Mortierella (−0.9789, −0.9580); Talaromyces was highly positively correlated with Fusarium (0.9720); Botryotrichum was highly positively correlated with Nigrospora (0.9720); Thermomyces was highly positively correlated with Xerochrysium (0.9650); Xerochrysium and Thermomyces were highly positively correlated with Botryotrichum (0.9510, 0.9510); and Arthrographis was highly positively correlated with Mortierella and Gibellulopsis (0.9507, 0.9507). The correlations among fungi were stronger than those among bacteria.

Figure 5 (A, B) Correlation analysis among different microorganisms.

Correlation analysis between bacteria and fungi

Spearman correlation analysis was used to establish correlations between the bacteria and fungi in the fruit surface samples. At the phylum level, Ascomycota was positively correlated with Chloroflexi (P < 0.05; Fig. 6A). At the genus level, Fusicolla and Talaromyces were positively correlated with Escherichia_Shigella (P < 0.05), unclassified_Didymellaceae was significantly positively correlated with Massilia (P < 0.01) and Pantoea (P < 0.001), Alternaria was significantly positively correlated with Massilia and Pantoea (P < 0.05), and Filobasidium was significantly positively correlated with Massilia (P < 0.05) and Pantoea (P < 0.01; Fig. 6B).

Figure 6 (A, B) Correlation analysis between bacteria and fungi.

Prediction of bacterial and fungal functions

The Picrust2 software was used to predict the functions of bacteria with a relative abundance >1%. The most important functions of the bacteria from all the samples were metabolism (77.96 ± 0.60%), environmental information processing (6.85 ± 0.56%), and genetic information processing (6.73 ± 0.37%; Fig. 7A). FUNGuild was used to predict the functions of fungi with a relative abundance >1%. The most important functions of the fungi from all the samples were undefined saprotrophs (46.78 ± 5.71%) and plant pathogen (19.93 ± 5.04%; Fig. 7B).

Figure 7 Prediction of bacterial and fungal functions.

(A) The functions of bacteria predicted by Picrust2 software; (B) the functions of fungi predicted by FUNGuild.

Discussion

Studies have shown that, as with other trees, microbial diversity on the surface of fruit trees is influenced by many factors, including management practices, genotype, soil properties and rootstock (Abdelfattah et al., 2020; Liu et al., 2018; Wassermann, Müller & Berg, 2019; Cui et al., 2021). However, research on microbial diversity on the surface of post-harvest fruits is still very limited, and the studies that have been conducted have mainly focused on the effect of geographic factors on the microbial diversity of post-harvest fruit surfaces (Abdelfattah et al., 2021). Current data on the characterization and relationships of epidermal microbial communities among different fruits from the same planting area are also insufficient. In the present study, four varieties of fruit with different genetic relationships (XHPG, YHL, YUL, and BTL) were taken as samples from the same plantation in northwest China. The community structure characteristics of surface microorganisms (including bacteria and fungi) were analyzed to reveal the distribution pattern of surface microbes on fruits with different genetic relationships of the same geographic origin. The results showed that there was no significant difference in bacterial diversity on the surfaces of different fruit varieties, but there were significant differences in fungal diversity. This suggests that fruit variety more significantly affects fungal diversity than bacterial diversity. Beta diversity analysis results showed that the composition of both bacteria and fungi on the surfaces of the four fruit varieties exhibited significant species specificity. Clustering analysis results indicated that the bacterial and fungal compositions of the three varieties of plums were similar, implying that fruit type is a key factor influencing the microbial composition of the fruit surface.

The most abundant bacterial phylum on the surface of all four fruit samples was Proteobacteria, followed by Bacteroidota, Firmicutes, Actinobacteriota, and Acidobacteriota. This finding aligns with the microbial diversity previously found on apple surfaces and common grape surfaces collected globally, indicating that these bacteria, especially Proteobacteria, are predominant on fruit surfaces (Abdelfattah et al., 2021). However, Zhang et al. (2022) found that the main phylum of apple surface bacteria was Actinobacteria (Zhang et al., 2022), which differs from the findings of the present study. This difference may be due to geographical factors, as different regions can have different microorganisms on the surface of the same variety of fruit. In the present study, at the genus level, the dominant bacteria included unclassified_Bacteria, Sphingomonas, unclassified_Cyanobacteriales, Rothia, Massilia, Pantoea, Pseudomonas, unclassified_Gemmatimonadaceae, Hymenobacter, and Escherichia_Shigella. Sphingomonas and Pseudomonas were also the most abundant genera found on apple surfaces in previous studies (Abdelfattah et al., 2021), differing from the common fungal genera found on grape surfaces.

The phylum Proteobacteria is associated with plant nitrogen utilization; an increase in nitrogen leads to a higher abundance of Proteobacteria and a decrease in Acidobacteria (Zhang, 2018). The abundance of Proteobacteria found on the surface of apples globally and on wine grapes from Nyingchi, Tibet can reach as high as 60–70% (Abdelfattah et al., 2021; He et al., 2022). In southeastern Tibet, the abundance of Proteobacteria on apple surfaces was 28.07% (Zhang et al., 2022). In this study, the abundance of Proteobacteria on the surface of four varieties of fruit was 40.80%, which differs from previous research. This discrepancy may be due to differences in the study regions, indicating that geographical factors and soil element content are also key factors influencing microbiota on fruit surfaces. Most bacteria in the phylum Firmicutes are fermentative bacteria, while the phylum Bacteroidetes includes various bacteria with functions such as carbohydrate fermentation (Yang et al., 2020). Sphingomonas is a dominant genus in soil (Zhang et al., 2018a), and Pseudomonas is widely distributed in plants, playing beneficial roles in plant growth, biological control, and nutrient utilization (Zhao et al., 2021; Lee et al., 2019; Lu et al., 2020). Therefore, the bacterial communities on the fruit surfaces in the present study were rich in microorganisms derived from the soil and in microorganisms related to plant growth and disease prevention. Meanwhile, most of the bacteria in Myxococcota phylum can eliminate plant pathogens (Bull, Shetty & Subbarao, 2002), results in our study indicated that the abundance of Myxococcota in the BTL sample was significantly higher than in the other three groups, this maybe the reason that BTL had a longer storage period than the other plum varieties.

On the surfaces of the four varieties of fruit in the present study, the most abundant fungal phyla were Ascomycota and Basidiomycota, which is consistent with the most abundant fungal phyla found on apple and grape surfaces in previous studies (Abdelfattah et al., 2021; He et al., 2022). At the genus level, the most abundant fungi found in the present study were Mortierella, Fusarium, unclassified_Fungi, unclassified_Didymellaceae, Aspergillus, Cladosporium, Alternaria, Filobasidium, Fusicolla, and Talaromyces. According to previous reports, Aspergillus and Cladosporium are typical microorganisms on grape surfaces, (Abdelfattah et al., 2021), and Alternaria, Mortierella, Fusarium, Cladosporium, and Filobasidium are common microorganisms found on apple surfaces (He et al., 2022). Among the high abundance microorganisms found on fruit surfaces in the present study, Mortierella is an important probiotic genus in the rhizosphere soil of apples (Zhang et al., 2022), and Cladosporium, Alternaria, and Fusarium are fungi with high isolation frequency in apple peels (Li et al., 2016). Notably, Cladosporium play an important role in plants (Bai et al., 2020; Zhou, Tang & Guo, 2018); Cladosporium rots grapes and produces toxins that alters the chemical composition of grapes and affects yeast growth during wine fermentation (He et al., 2022). Alternaria can produce more than seventy kinds of toxic fungal metabolites such as streptozotocin, streptozotocin monomethyl ether, and streptozotocin ketoacid, causing grape black spot disease; ingestion of these mycotoxins by humans and animals can lead to acute or chronic poisoning, with some of these mycotoxins being carcinogenic, teratogenic, or mutagenic (Motta & Valente Soares, 2001; Li, 2001). These results indicate that fungal and bacterial populations on fruit surfaces are similar as they are primarily derived from soil and are related to plant growth and fruit decay.

A correlation analysis in the present study revealed significant associations between fungi and fungi, bacteria and bacteria, as well as fungi and bacteria on the surfaces of the four fruit varieties. The associations among fungi were greater than those among bacteria, indicating symbiotic and interactive relationships among the surface microbiota, which is in line with the findings of previous studies (Zhang et al., 2022). Furthermore, the dominant fungi and dominant bacteria were positively correlated, Ascomycota have been known to produce a wide range of antimicrobial agents, which can be advantageous to the protection of plants against pathogens (Jibola-Shittu et al., 2024). Chloroflexi phylum is a denitrifying bacterium that can also inhibit pathogenic bacteria (Schwartz et al., 2022). In tea garden soil, Chloroflexi is a common bacterial phylum, while Ascomycota is a common fungal phylum (Jibola-Shittu et al., 2024), indicating that these two microorganisms have a symbiotic relationship, which is consistent with the results of our study. But the specific interaction mechanisms need to be further explored. Meanwhile, in the present study only four fruit varieties were studied, and the sample size was small, so further study need to be performed with large sample size to supply more information of the microorganisms diversity on the surface of fruit.

Conclusions

This study collected four varieties of fruits from the same orchard in Donghe District, Baotou City, Inner Mongolia, and analyzed their peel microbiota diversity and composition using high-throughput sequencing of 16S rRNA and ITS. The results showed no significant difference in bacterial diversity on the surfaces of these fruits, but did show significant differences in fungal diversity. The most abundant bacterial phyla on the surfaces of these fruits were Proteobacteria, Bacteroidota, and Firmicutes; the most abundant fungal phyla were Ascomycota, Basidiomycota, and Mortierellomycota. The microbial composition of epidermal microorganisms varied between the different fruits. However, the microbial compositions of the three plums were relatively similar, indicating that fruit type is a key factor affecting peel microbial composition. There are some differences between the most abundant microbial species in the present study and those reported on fruits in other areas, suggesting that geographical factors also influence fruit peel microbiota composition. The microbial correlation analysis revealed significant correlations between microorganisms with the highest abundance, suggesting the existence of symbiosis and mutualism among fruit epidermal microorganisms in the present study, but the specific mechanisms need to be further explored. This study also has limitations, as only four fruit varieties were studied, and the sample size was small. Future studies should expand the number of fruit varieties included and increase the sample size of fruits studied. However, the results of this study can still be used as a reference for the surface microbial composition and diversity of the four fruits included and provide some reference for post-harvest fruit preservation.

Abbreviations

YHL Yu Emperor Plum

BTL Crystal Sugar Plum

YUL Yu Plum

OTUs operational taxonomic units

NMDS non-metric multidimensional scaling

LDA linear discriminant analysis

LEfSe linear discriminant analysis (LDA) effect size

PCR polymerase chain reaction.

Additional Information and Declarations

Competing Interests

Author Contributions

Data Availability

The authors declare that they have no competing interests.

Shan He conceived and designed the experiments, performed the experiments, prepared figures and/or tables, and approved the final draft.

Li Gao conceived and designed the experiments, performed the experiments, analyzed the data, prepared figures and/or tables, authored or reviewed drafts of the article, and approved the final draft.

Zhuomin Zhang analyzed the data, prepared figures and/or tables, and approved the final draft.

Zhihui Ming analyzed the data, authored or reviewed drafts of the article, and approved the final draft.

Fang Gao performed the experiments, authored or reviewed drafts of the article, and approved the final draft.

Shuyi Ma performed the experiments, analyzed the data, authored or reviewed drafts of the article, and approved the final draft.

Mingxin Zou analyzed the data, authored or reviewed drafts of the article, and approved the final draft.

The following information was supplied regarding data availability:

The raw sequence reads are available at NCBI Bioproject: PRJNA1131979 (Bacteria, SAMN42328755–SAMN42328766) and PRJNA1131983 (Fungi, SAMN42328777–SAMN42328788).

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
