# Peer review of "Diversity analysis of microorganisms on the surface of four summer fruit varieties in Baotou, Inner Mongolia, China"

_PeerJ, doi:10.7717/peerj.18752_

## Round 0.1 · original submission · Major Revisions

While the reviewers generally felt that the manuscript was well written, all three reviewers had concerns about the manuscript, with the level of detail included in the background information in the introduction, description of the methods and small sample sizes all being repeatedly mentioned.

Reviewer 1 ·

Basic reporting

In the literature section, the subject of microbial preservatives needs to be added more and reviewed and referred from previous studies.

In the materials and methods section, the expected PCR product size needs to be included and well-described.

Results sections were well-described but some numbers need to be written in the word.

Figures and diagrams were labeled properly and well-described.

This paper was well-written and organized properly.

Experimental design

Methods were described well but need some minor correction.

Validity of the findings

All data were clear and well explained.

Additional comments

This paper need some correction as highlight in the reviewed paper.

Annotated reviews are not available for download in order to protect the identity of reviewers who chose to remain anonymous.

Reviewer 2 ·

Basic reporting

This study analyzed the microbial diversity on the surface of four summer fruits from Baotou, Inner Mongolia, using high-throughput sequencing. Here are my major comments.

1. There are several language problems in the manuscript. For example:
L24: "displayed no notably difference" should be "displayed no notable difference."
L43: "in turn leads to a substantial loss" should be "which in turn leads to substantial losses."
2. Introduction Needs More Detail: The introduction doesn’t cover the background thoroughly. You should add more information about studies on microbial diversity on fruit surfaces in different regions and identify the research gap.
3. There are only 6 citations in the whole Introduction. That is not enough.
Also, the latest citation in Introduction is in 2018, which cannot provide the current progress of this area.
4. L68-69: “Additionally, the diversity of surface microorganisms varies among fruits of different kinds, origins, and species, leading to different decay-causing bacteria.” Sure, the surface microorganisms varies due to many reasons. But why it leading to different decay-causing bacteria? As we know, pathogenic fungi causing much more diseases than bacteria during postharvest stage.
5. The author should provide more information about the fruit used in this study. That is very important because it directly impact how you design your experiment. And why you choose these four fruits.

Experimental design

6. L91-92: “In the lab, fruit surfaces were collected using sterilized surgical instruments, vortexed thoroughly, and stored at -80°C.” Please describe this step in detail. What surgical instruments did you use? How could you be sure it has been vortexed thoroughly? How long?
7. L94: which kit did you use?
8. What programs did you use for the PCR?
9: L105: How about the fungal annotation?
10: L110: Which software did you use for beta diversity? What parameters or calculations did you use?
11: How did you perform the visualization for the Fig.5 and Fig.6?

12. Some figures are hard to understand because the titles and labels are not clear enough.

13. L226: bacteria and bacteria?
14. L227: fruit skin?
15: L229 and L232: How to define highly positively correlated?

Validity of the findings

16: L250-251: How the beta diversity could show the microbial composition exhibits significant species specificity? Also, here is a grammar error: exhibited.
17. The study doesn’t talk enough about the importance of geographical factors on microbial communities. This should be included in the discussion.
18. Incomplete Conclusions: The conclusion section is missing a discussion on the limitations of the study. You only studied four types of fruit, which is a small sample size. This limitation should be mentioned in the conclusion.
19: The author emphasized this study provided reference for the post-harvest fruit preservation technology, which has exaggerated results. It can be said that the results of this study provide a reference for the surface microbial composition and diversity of these four fruits. However, it is too far to provide a reference for preservation technology. There is no experiment on postharvest preservation technology in this study.

Additional comments

no comment

Reviewer 3 ·

Basic reporting

The authors in the study entitled “Diversity analysis of microorganisms on the surface of four summer fruits in Baotou, Inner Mongolia, China” study the surface microbiota of 3 plum fruit cultivars and apple using ITS and 16S sequencing. The authors explored the alpha-, beta-diversity of these communities, revealed abundant taxa and conducted a correlation analysis between the microbial taxa. Although the analysis we correctly done and presented, I have some reserve regarding the study that I list below.
• The number of replicates is too small for the conclusion drawn. The authors should include more replicates per sample, as it is not clear so far whether these are one time observations or more general.
• The study catalogued very nice descriptive analysis, but it was missing hypothesis driven testing and their biological significance regarding for instance fruit decay, or fruit pathogens. Also, it is unclear why the authors included apples to the set and we did not see the link or the authors discussed its relevance.
• There are several instances for lack of precision and detail that I highlight below.
L23, What are these fruit or related plant species and similar in L25.
L26, These phyla are known plant associated bacteria, it would be more informative to indicate if new genus or species are detected and what they are.
L28, not clear by ‘Through’ in the text
L33-35, The authors could be more specific and indicate the taxa and provide clear hypothesis
Intorduction the introduction reads good, but the authors could provide more background about fruit microbiota studies, what bacteria and fungi colonize these fruits.
L92, This part could be further detailed, particularly the collection of the microbiota
L94, what as the kit or method for the extraction
L95, how the authors resolve the issue of plant and plastid associated reads?
L100, which kit version and method were used in to sequence.
L102, raw data were not provided and could not check the analysis.
L119, The authors did not provide the version of each package or software and there is no link to raw data or code that generated these analysis!
L131, the venn comparison is informative, but the authors should tell more about what are these taxa or the most dominant ones
L144, The authors should perform PERMANOVA text to support thier claim
Figure2B, the panel are not well described in the text
L153, This is a nice analysis, but the authors could take advantage and compared between the locations and cultivars
L177 and L203, The authors could use Picurst and funguild to predict functions of these microbiota and draw hypothesis

Experimental design

Indicated in box 1

Validity of the findings

Indicated in box 1

---

## Round 0.2 · Major Revisions

While the reviewer acknowledges that the authors have improved the manuscript, they still have concerns about several aspects. In addition, may I offer the suggestions below:

1. Please provide the versions of the Silva and UNITE databases used.
2. Lines 145-146. Please provide versions and references for Picrust2 and FUNGuild.
3. I agree with the reviewer that more could be made of the biological importance of the presence of some of these taxa, as you have done with Proteobacteria and nitrogen utilization. Is there anything you could indicate for another two or three taxa?

Reviewer 3 ·

Basic reporting

I thank the authors for making improvements to the text and figures. The text reads better and figures improved. I, unfortunately, could not see an improvement into biological conclusions from this survey study. The authors listed their findings and did not provide biological significance in the different sub-sections of the results. For instance, the authors indicate that microbes could extend the shelf-life of fruits by delving into the details. Not to discourage, the authors could indicate if one of the plum varieties is better preserved than another; what we learn from the apple microbiome is applying the apple microbiome of either plum would affect or deteriorate the fruit. As is, the ms lack substance on the biological findings.
Other comments
L138, which of the distances, Jaccard or unweighted Unifra were used? If both, it should be indicated both.
Fig 1B and E the numbers are too small to see and the authors should indicate if these shared OTUs were conserved in across all replicates.
Fig6 provide nice results, the authors should discuss it more in the text and provide example if they exisit from literature about negatively co-occurring taxa.

Experimental design

The replicates remain low and the authors might acknowledge it in the discussion.

Validity of the findings

No comments

---

## Round 0.3 · Minor Revisions

The authors have made an effort to address comments from the previous reviewer and myself. However, several of these contain significant errors:

1. The reference for Picrust2 is Douglas, G.M., Maffei, V.J., Zaneveld, J.R. et al. PICRUSt2 for prediction of metagenome functions. Nat Biotechnol 38, 685–688 (2020). https://doi.org/10.1038/s41587-020-0548-6 and not Liu et al (that paper doesn't even use Picrust).
2. The reference for FUNGuild is Nguyen NH, Song Z, Bates ST, Branco, S, Tedersoo L, Menke J, Schilling JS, Kennedy PG. 2016. FUNGuild: an open annotation tool for parsing fungal community datasets by ecological guild. Fungal Ecology 20:241-248. and not Gao et al.

The data availability statement points to raw reads for a study of the gut microbiome of relict gulls (which was what the Liu et al study above was on). These need to be corrected for the correct raw data for this study.

Line 323: what do the authors mean by "crack plant pathogens"? I would suggest re-wording this.

---

## Round 0.4 · accepted · Accept

The authors have addressed the majority of comments, and I am happy with the current version.